# Influence of Dietary Polyunsaturated Fatty Acid Intake on Potential Lipid Metabolite Diagnostic Markers in Renal Cell Carcinoma: A Case-Control Study

**DOI:** 10.3390/nu16091265

**Published:** 2024-04-24

**Authors:** Yeon-Hee Kim, Jin-Soo Chung, Hyung-Ho Lee, Jin-Hee Park, Mi-Kyung Kim

**Affiliations:** 1Cancer Epidemiology Branch, Division of Cancer Epidemiology and Prevention, National Cancer Center, 323 Ilsandong-gu, Goyang-si 10408, Republic of Korea; yh0227@ncc.re.kr (Y.-H.K.); jhblue99@ncc.re.kr (J.-H.P.); 2Department of Urology, Center for Urologic Cancer, Research Institute, Hospital of National Cancer Center, 323 Ilsandong-gu, Goyang-si 10408, Republic of Korea; cjs5225@ncc.re.kr (J.-S.C.); uroh@ncc.re.kr (H.-H.L.)

**Keywords:** diagnosis markers, carnitine palmitoyltransferase 1, lysophosphatidylcholine, metabolite, polyunsaturated fatty acid, renal cell carcinoma

## Abstract

Non-invasive diagnostics are crucial for the timely detection of renal cell carcinoma (RCC), significantly improving survival rates. Despite advancements, specific lipid markers for RCC remain unidentified. We aimed to discover and validate potent plasma markers and their association with dietary fats. Using lipid metabolite quantification, machine-learning algorithms, and marker validation, we identified RCC diagnostic markers in studies involving 60 RCC and 167 healthy controls (HC), as well as 27 RCC and 74 HC, by analyzing their correlation with dietary fats. RCC was associated with altered metabolism in amino acids, glycerophospholipids, and glutathione. We validated seven markers (l-tryptophan, various lysophosphatidylcholines [LysoPCs], decanoylcarnitine, and l-glutamic acid), achieving a 96.9% AUC, effectively distinguishing RCC from HC. Decreased decanoylcarnitine, due to reduced carnitine palmitoyltransferase 1 (CPT1) activity, was identified as affecting RCC risk. High intake of polyunsaturated fatty acids (PUFAs) was negatively correlated with LysoPC (18:1) and LysoPC (18:2), influencing RCC risk. We validated seven potential markers for RCC diagnosis, highlighting the influence of high PUFA intake on LysoPC levels and its impact on RCC occurrence via CPT1 downregulation. These insights support the efficient and accurate diagnosis of RCC, thereby facilitating risk mitigation and improving patient outcomes.

## 1. Introduction

Globally, renal cell carcinoma (RCC) accounts for 70–80% of kidney cancers and is a highly aggressive subtype [1,2]. Owing to the asymptomatic early stages, late diagnosis and high metastasis rates are common [3]. Timely stage-1 RCC detection enhances 5-year survival by 7.25-fold compared to that at stage-4 RCC detection (12%) [4], reducing disease progression, recurrence, and metastasis. RCC diagnosis via pathology or surgery is challenging for small tumors [5]; therefore, non-invasive and specific biomarkers for RCC detection are urgently required.

To the best of our knowledge, 13 prior case-control studies have identified potential metabolic biomarkers in plasma [6,7,8,9,10] or serum [11,12,13,14,15,16,17,18] that distinguish RCC from controls. The proposed profiles include lipids, lipoproteins [6,7,8,9,13,14,15,17,18], amino acids [6,7,8,10,12,13,14,17], and carbohydrates [7,11,13,14,16]. However, robust RCC diagnostic biomarkers remain unclear.

Abnormal RCC growth alters glucose, lipid, and amino acid metabolism [2]. Clear-cell RCC is driven by the activation of hypoxia-inducible factors (HIFs) derived from the von Hippel–Lindau (VHL) mutation, which accelerates tumor progression [19,20]. Increased HIF levels in RCC cause lipid-droplet accumulation and elevated levels of free fatty acids (FFAs), particularly polyunsaturated fatty acids (PUFAs), which react with reactive oxygen species (ROS) to cause lipid peroxidation [21]. Amino acid metabolism critically regulates RCC stages, thereby affecting growth, angiogenesis, and prognosis [22].

Amino acid and lipid metabolism are vital in RCC. Prior studies lacked systematic analyses of specific amino acids and lipids in the plasma of patients with RCC, as they mainly used qualitative mass spectrometry methods [23], such as proton nuclear magnetic resonance [7,13,16], MS or multivariate models [10,11,12,14,18], and liquid chromatography (LC)-MS [8,9,15,17]. Our study quantified plasma metabolites and lipids using an AbsoluteIDQ p400 high-resolution (HR) kit and ultra-high-performance LC (UHPLC)-MS, which allowed for absolute quantification. Blood-based samples are less variable, diet-sensitive, and rich in lipids, which are crucial for biology [8]. Machine-learning (ML) analysis of extensive cancer data has shown promise in determining prognosis [24]. We aim to employ ML algorithms to identify predictive plasma markers with accuracy.

Risk factors for RCC include smoking, obesity, and hypertension, but dietary links are limited [25]. Previous studies on RCC show varied results regarding fat intake [26,27,28,29]. In a European investigation, fish intake was found to have no association with an increased risk of RCC [29], whereas a study conducted in Japan indicated otherwise [28]. Despite the significance of lipids in RCC, previous studies have yielded varying results. Specifically, studies linking nutrients to blood metabolites are rare.

Therefore, herein, we aim to identify robust potential markers for the diagnosis of RCC through the quantification of plasma metabolites, machine learning, and marker validation. Furthermore, we investigated the correlations and mechanisms between the potential markers, dietary fats, and food to elucidate their impacts on RCC development.

## 2. Materials and Methods

### 2.1. Study Participants

In this research, we included adults (19+ years) diagnosed with RCC at the National Cancer Center, South Korea. Two patients with RCC were excluded due to lung cancer metastasis. A healthy control (HC) group without cancer was also gathered from the same center. We randomly divided RCC and HC samples into discovery and validation sets at a ratio of 70:30 (discovery set: 60 RCC and 167 HC, validation set: 27 RCC and 74 HC). A questionnaire assessed age, gender, height, weight, alcohol use, and smoking. Body mass index (BMI) was measured by taking weight (kg) divided by height (m)^2^. Smoking/drinking history was noted if present. Carcinomas were categorized into four types (clear cell, chromophobe, papillary, and unclassified), and the categories included cancer site, T stage, pathology stage, surgery type, nuclear grade, tumor size, margin, necrosis, and invasion presence. This study was approved by the National Cancer Center Institutional Review Board (IRB No. 2019-0116, approval date of ethical statement: 3 June 2019). Consent was attained per the Declaration of Helsinki.

### 2.2. Plasma Sample Preparation

Participants fasted for 12 h prior to blood collection. Blood was drawn into K2 EDTA tubes (BD Vacutainer, BD Biosciences, Franklin Lakes, NJ, USA), centrifuged at 3000 rpm for 20 min at 4 °C to get plasma, and stored at −80 °C for later analysis.

### 2.3. UPLC/Orbitrap MS and Metabolite Quantification Analysis

We used Vanquish Flex UHPLC (Thermo Fisher Scientific, Waltham, MA, USA) connected to Q ExactiveTM Hybrid Quadrupole-Orbitrap MS (Thermo Fisher Scientific) for targeted metabolic and lipid profiling. We employed the AbsoluteIDQ p400HR kit (Biocrates, Innsbruck, Austria) to analyze 408 metabolites and lipids in the blood, including 21 amino acids, 21 biogenic amines, 1 hexose, 172 phosphatidylcholines, 24 lysophosphatidylcholines, 31 sphingomyelins, 9 ceramides, 55 acylcarnitines, 14 cholesteryl esters, 18 diglycerides, and 42 triglycerides. The analysis was performed following the kit’s instructions. Quantities were calculated using software, including Quant Browser (Thermo Xcalibur 4.3.73.11) and MetIDQ (Oxygen-DB110-3023), based on obtained spectra from the instrument.

### 2.4. Metabolomic Data Processing

Ninety-one metabolites with values < LoD or <LLoQ were excluded using MetaboAnalyst 5.0 (https://www.metaboanalyst.ca/ (accessed on 18 April 2023)). Missing values were replaced with an LoD value (1/5 of minimum peak intensity). Normalization used log transformation and auto-scaling, analyzing 284 metabolites. Principal component analysis and partial least squares discriminant analysis (PLS-DA) were conducted using MetaboAnalyst. For PLS-DA, R2 > 0.5 and Q2 > 0.5 were used for goodness of fit. Differential metabolites were selected with an FDR-adjusted *p* < 0.05 and multivariate PLS-DA variable importance in projection (VIP) > 1.0. Heatmaps and volcano plots were generated to identify differentially expressed metabolites.

### 2.5. Identification of RCC Diagnostic Candidate Markers through Machine Learning

In this study, we employed ML to analyze the blood metabolomes of patients with RCC using a case-control design. An optimal RCC diagnostic model was developed using the GOSS algorithm paired with lightweight GBM. The model’s performance was assessed via 5-fold cross-validation on a training set. The model’s efficiency and generalizability were evaluated with a fixed test set, ensuring reliable estimates. After trying XGBoost, LightGBM, GBM, and Random Forest, Random Forest was chosen as the best-performing. The model’s performance was comprehensively evaluated using 5-fold average scores for metrics such as accuracy, AUC, sensitivity, specificity, F1 score, and precision.

### 2.6. Receiver Operating Characteristic (ROC) Curve, Enrichment, and Pathway Analysis

The ROC curve analysis identified potential RCC-associated markers. MetaboAnalyst performed classical univariate and multivariate ROC analyses. The ROC was used to assess metabolite-marker performance. ROCs compared sensitivity and specificity for dichotomous outcomes. The area under curve (AUC) was used to assess marker effectiveness. Both analyses used cutoffs for markers with high sensitivities and specificities (AUC > 0.80). The linear SVM (Support Vector Machine) algorithm was used for ROC curve analysis. Log2 fold change (FC) shows the metabolite differences between groups. Enriched lipid–metabolite sets used a *p* < 0.05 cutoff; pathways used a *p* < 0.05 FDR cutoff with MetaboAnalyst and Cytoscape (3.10.0, https://cytoscape.org/ (accessed on 20 April 2023)) to visualize major metabolite networks, and the shape sizes showed significance.

### 2.7. Carnitine Palmitoyltransferase (CPT) Family Genes Expression Analysis and CPT1 Measurement

We analyzed CPT family genes (CPT1A, CPT1B, and CPT2), carnitine-acylcarnitine translocase (CACT), and carnitine acetyltransferase (CrAT) in human kidneys using published GEO microarray datasets (GSE781 and GSE6344) via the GEO database (https://www.ncbi.nlm.nih.gov/geo/ (accessed on 10 June 2023)). The data were compared based on relative expressions.

CPT1 levels were measured using a CPT1 enzyme-linked immunosorbent assay (ELISA) kit (mbs724213, MybioSource, Inc., San Diego, CA, USA). Plasma samples (1:5 dilution) were added to 96-well plates with standards. After adding a conjugate, the plates were incubated at 37 °C for 1 h and then washed five times. Substrates A and B (50 μL each) were added to the wells and incubated in the dark for 10 min. The reaction was stopped with a 50 μL stop solution. The optical density was measured at 450 nm using a UV–vis spectrometer (SPECTROstar Nano, BMG Labtech, Ortenberg, Germany). CPT1 concentrations were determined by comparing the standard curve with the absorbance values of the samples at 450 nm.

### 2.8. Dietary Fat and Nutrient Intake Analysis and Correlations with Metabolites

The nutrition of each participant was assessed with a semi-quantitative Food Frequency Questionnaire (FFQ), comprising 95 items. The FFQ surveyed 219 HCs and 27 individuals with RCC. FFQ data were analyzed via the Computer-Aided Nutritional Analysis Program 5.0 (CAN-pro 5.0, http://canpro5.kns.or.kr/ (accessed on 11 May 2023)). We computed the daily averages for energy, carbohydrates, lipids, and amino acids. Nutrient-adequacy ratios were assessed based on the 2020 Dietary Reference for Koreans provided by the Ministry of Health and Welfare and the Korean Nutrition Society. The intake of dietary fats or nutrients was calculated as the amount consumed per day relative to a daily average of 1000 kcal. Spearman’s rank correlation coefficient was used to analyze the metabolite concentration correlation with nutrients and food groups.

### 2.9. Statistical Analysis

We employed SPSS (version 26.0, Chicago, IL, USA) for statistical analyses of the general characteristics using the chi-squared test, student’s *t*-test, and Fisher’s exact test. The results were presented as mean (SD) or *n* (%), with *p* < 0.05 indicating significance. Python (version 3.9.12, Wilmington, DE, USA) performed a multivariate logistic regression and correlation analysis for metabolites and nutrients. Metabolomics data used an FDR-adjusted *p* to identify significant compounds in the HCs vs. RCC group. Multivariate PLS-DA VIP scores > 1.0 were assessed with MetaboAnalyst 5.0.

## 3. Results

### 3.1. Clinical and Diagnostic Characteristics of HC and RCC Groups

The clinical and diagnostic characteristics of the HC and RCC groups are summarized in Table 1. Both the RCC and HC groups in both sets had an average age in their 60s, with a higher proportion of males, although the difference was not significant. In both sets, there were a notable number of individuals with a BMI classified as overweight or obese (discovery set: HC 72.6%, RCC 71.7%; validation set: HC 70.2%, RCC 85.2%).

In the validation set, the RCC prevalence was slightly higher compared to HC. Smoking rates were similar between groups. The HC group had 20–30% more alcohol experience than the RCC group. Stage 1 was the most common stage for RCC in both sets (discovery set 78.4%, validation set 77.8%) (Appendix A). There was no lymph node involvement. Radical nephrectomy accounted for approximately 20%, with the remainder being partial nephrectomy. The Fuhrman nuclear grade was predominantly 2/3. The volume of the RCC tumors was measured at 25.2 ± 15.5 cm^3^.

### 3.2. Metabolomic Profiling for Identification of Candidate Metabolites

Metabolomic profiling was conducted to identify candidate metabolites, as depicted in Figure 1A. A non-targeted metabolomics analysis was performed on plasma samples obtained from 167 HC and 60 RCC individuals, along with 35 QC samples. After excluding metabolites, normalizing, model validating, and applying cutoffs via PLS-DA VIP values and FDR-adjusted *p* values, 79 metabolites remained for analysis (Appendix A). PLS-DA plots displayed intergroup variations (Appendix A). VIP values from PLS-DA indicated potential discriminatory metabolites (VIP > 1.0). The goodness of fit and predictive ability of the model were estimated by the R2 and Q2 values (0.845 and 0.624), clearly discriminating groups (Appendix A). A heatmap based on 79 differential metabolites (VIP > 1.0, FDR *p* < 0.05) showed substantial HC-RCC metabolite profile differences (Figure 1B). Among them, 74 metabolites (e.g., decanoylcarnitine, LysoPC (16:0)) were downregulated in RCC vs. HCs, whereas 5 metabolites (e.g., l-glutamic acid) were upregulated in RCC vs. HCs (Figure 1C, Appendix A).

### 3.3. Quantitative Analysis and Multivariate Logistic Regression Analysis Results for Candidate Metabolites

The quantitative analysis results for candidate metabolites are as follows (Figure 2A–G). Among the key metabolites, l-Glutamic acid increased in RCC, while decanoylcarnitine, l-Tryptophan, LysoPC (16:0), LysoPC (18:0), LysoPC (18:1), and LysoPC (18:2) decreased. After controlling for age, sex, BMI, smoking, and drinking, we calculated the odds ratios (ORs) for key metabolites (Figure 2H, Appendix A). Six of the metabolites, except l-Glutamic acid, consistently exhibited a negative OR.

### 3.4. ML for RCC Diagnostic Candidate Metabolites Prediction

Upon comparing the performance metrics of four models (XGBoost, LightGBM, GBM, and Random Forest), Random Forest emerged as the optimal model (Appendix A). Our selected model demonstrated an impressive 95.9% AUC, indicating robust discrimination between RCC and HC (Figure 2K,L). High sensitivity (0.9556 ± 0.0465) and specificity (0.8720 ± 0.0642) were achieved. Further analyses encompassed the top-20 metabolites of importance, ROC curve, confusion matrix, POD label, and SHAP techniques (Figure 2I,J,M and Appendix A). Building upon the previously analyzed VIP, FDR *p*-value, logistic regression *p*-value, and the top-20 metabolites from ML, seven were selected as diagnostic candidate markers (l-Tryptophan, LysoPC (16:0), LysoPC (18:0), LysoPC (18:2), decanoylcarnitine, LysoPC (18:1), and l-Glutamic acid).

### 3.5. Discovery and Validation of Seven Potential RCC Diagnostic Markers

We evaluated whether the previously selected seven candidate markers could effectively distinguish RCC from HC using an ROC analysis (Appendix A). All seven candidate markers exhibited AUC values > 0.8 and *p* < 0.05 in discovery (Figure 3A–G). In the validation set, these markers were confirmed as potential diagnostic markers for RCC, each distinguished with an AUC of over 0.78. Further analysis explored combinations of these metabolites (Figure 3H,I), with ROC analysis demonstrating high AUC values of 0.972 in the discovery set and 0.969 in the validation set for the panel of seven markers. This suggests that these selected markers could serve as potent potential biomarkers for the diagnosis of RCC. Our analysis was further conducted to evaluate the utility of these markers in detecting early-stage versus late-stage RCC. The ROC analysis for five to seven markers indicated that the AUC values for differentiating HC vs. T stage 1 were slightly higher than those for HC vs. T stages 2 and 3 (Figure 3J). Additionally, after adjusting for age, sex, BMI, smoking, and alcohol consumption, the direction of the ORs for each of the seven markers between HC and T stage 1 remained consistent with the direction of the ORs between HC and all RCC stages (Appendix A).

### 3.6. Enrichment and Pathway Analyses of HCs and RCC Differential Metabolites

To explain RCC-related metabolic function, we analyzed the enrichment of metabolites in all RCCs and HCs (Appendix A). Fatty acyls, including decanoyl carnitine, emerged as key lipids altered in RCC vs. HCs, and 43 glycerophospholipids, including LysoPC (16:0), significantly differed in RCC (*p* < 0.05). Human metabolic pathway enrichment indicated that RCC was linked to an amino acid (alanine, aspartate, glutamic acid, and tryptophan) and glycerophospholipid (LysoPCs) metabolism. Notable affected pathways included alanine, aspartate, and glutamate metabolism; arginine and proline metabolism; arginine biosynthesis; and histidine metabolism.

RCC-associated metabolic pathways analysis identified 35 pathways (Figure 4, Appendix A). Among the 11 significantly changed pathways, alanine, aspartate, and glutamate metabolism stood out (FDR-adjusted *p* = 3.80 × 10^−41^, pathway impact = 0.5345). Glycerophospholipid metabolism altered, impacting 24 phosphatidylcholines (PCs) and nine LysoPCs. Most PCs and LysoPCs decreased in RCC, except PC (15:1/22:2) and PC (21:0/22:2), which increased. Other pathways involved 12 amino acids, which are most notably depleted in RCC plasma, except l-aspartic acid and l-glutamic acid.

### 3.7. Decanoylcarnitine Decrease in RCC Is Regulated by CPT1a Downregulation

Prior research suggests that RCC increases HIF1 via VHL mutations, reducing CPT1a and fatty acid (FA) transport [30]. In order to elucidate the process by which decanoylcarnitine (acylcarnitine), a diagnostic marker for RCC, is regulated by intracellular CPT1, we validated the expression of CPT family members (CPT1A, CPT1B, CPT2, SLC25A20, and CrAT) using databases and ELISA (Figure 5). All tested CPT members were downregulated in RCC vs. HCs (Figure 5A–E). ELISA confirmed CPT1 reduction in RCC vs. HCs (Figure 5F).

### 3.8. Diagnostic Potential Marker Levels Are Affected by Clinical Factors

To further substantiate the clinical relevance of seven key markers, we analyzed changes based on clinical factors (RCC stage, site, age, BMI, and gender) (Figure 6). Decanoylcarnitine increased in stage 2 vs. stage 1, and LysoPC (18:0) decreased (Figure 6A). No site-based differences were observed (Figure 6B). With age rise, the RCC group showed a decreasing trend in LysoPC (18:0) and LysoPC (16:0) (Figure 6C). An increasing BMI raised L-glutamic acid in HCs and RCC, and LysoPC (18:2) dropped in RCC (Figure 6D). In HCs, females had lower l-glutamic acid, l-tryptophan, LysoPC (16:0), LysoPC (18:1), and LysoPC (18:2) than males. In RCC, l-tryptophan and LysoPC (18:2) decreased (Figure 6E). Some glycerophospholipids were lower in the blood of females and those with higher ages and BMIs.

### 3.9. High PUFA Consumption Is Associated with an Increased Risk of RCC Due to Its Correlation with Blood Fatty Acids

To examine the correlation between potential markers and dietary fats and food, a nutritional analysis was conducted (Appendix A). Higher PUFA, *n*-3 PUFA, and *n*-6 PUFA intake were correlated with an elevated risk of RCC (Figure 7A, Appendix A). The RCC group exhibited a higher per-calorie consumption of PUFAs than the HCs (Figure 7A). The average PUFA to recommended nutrient intake ratios were as follows: *n*-3 PUFAs, HC 83.2% and RCC 89.0%; and *n*-6 PUFAs, HC 88.9% and RCC 111.2%. Additionally, an increase in the consumption of fish and shellfish was associated with an elevated risk of RCC (Figure 7B). Correlation analysis revealed a negative association between LysoPC and PUFA, *n*-3 PUFA, fish, and shellfish (Figure 7C). This implies that PUFA consumption in RCC influences the reduction of LysoPC interacting with PC. Specifically, highly unsaturated FAs exhibited a stronger effect than saturated or monounsaturated FAs. Additionally, they exerted an influence on *n*-3 PUFA rather than *n*-6 PUFA (Appendix A). In contrast, LysoPC (18:0) demonstrated a positive correlation with CPT1.

## 4. Discussion

The aim of this study was to elucidate RCC diagnostic potential markers through the quantification of plasma metabolites, machine learning, and marker validation. Seven potential metabolites (l-glutamate, l-tryptophan, decanoylcarnitine, LysoPC (16:0), LysoPC (18:0), LysoPC (18:1), and LysoPC (18:2)) emerged as potential markers with a high AUC of 96.9% for distinguishing RCC in the validation set. We revealed correlations between these markers and dietary fat. These findings highlight the crucial role of lipid metabolism in RCC pathogenesis and introduce a new perspective on RCC diagnostics.

In elucidating the mechanism of RCC development, our findings suggest that high dietary intake of PUFAs induces changes in plasma PC and LysoPC levels, and the downregulation of CPT1, leading to decreased acylcarnitine levels, which likely impacts lipid synthesis and contributes to RCC onset. Initially, levels of LysoPC (16:0), LysoPC (18:0), LysoPC (18:1), and LysoPC (18:2) were reduced in RCC plasma compared to that in HCs. This observation is consistent with the findings of Lin et al., who reported a similar downregulation in RCC serum [17]. Multiple lysophosphatidylcholine (LPC) species decrease in cancer tissues, corresponding to PC species increase [31]. Our results also indicated an increase in PC (15:1/22:2) and PC (21:0/22:2), which contain omega-6 docosadienoic acid, in RCC. In RCC, LysoPC (18:1) and LysoPC (18:2) showed a negative correlation with *n*-3 PUFA, suggesting a strong impact of *n*-3 PUFA on polyunsaturated fat. Dietary PUFA, upon consumption, undergoes conversion into FAs and is further transformed into PC by acyl-CoA [32]. The conversion between LPC and PC is facilitated by LPC acyltransferase (LPCAT) and PLA2 [33]. LPCAT1 overexpression boosts LPC to PC conversion, influencing cell proliferation and fluidity and promoting cancer growth and metastasis via membrane PC-level alterations [31]. Short-chain LysoPCs exit the membrane and traverse into the cell, whereas long-chain FAs generated from the intake of dietary PUFAs enter the cell through FATP1 or CD36 [34]. Upon encountering carnitine via acyl-CoA, long-chain FAs form acylcarnitine and enter the mitochondria [35].

One of the seven potential markers, acylcarnitine, specifically decanoylcarnitine (C10 carnitine), was observed to decrease in RCC compared to HC, aligning with the study results of Liu et al. (*O*-decanoyl-l-carnitine, FC (RCC/HC) = 0.65) [9]. Currently, there is limited research on the association between RCC and decanoylcarnitine. Decanoylcarnitine is a medium-chain acylcarnitine [36]. Cells in different tissues require varying amounts of carnitine for survival [37]. Dysregulated levels of fatty acylcarnitines indicate impaired FA oxidation in the RCC group [38]. CPT1A, a key FA oxidation enzyme, showed reduced expression in RCC kidneys compared to normal kidneys in The Cancer Genome Atlas database [30,39]. Our mRNA expression analysis of CPT family members (CPT1A, CPT1B, CPT2, SLC25A20, and CrAT) from the GEO database confirmed downregulation in RCC compared to HCs. These findings align with our plasma CPT1 ELISA results. Moreover, HIF inhibits the direct HIF target gene CPT1, reducing FA translocation into mitochondria and promoting FA accumulation in lipid droplets, potentially contributing to RCC development [30]. These findings collectively support the notion of upregulated FA synthesis and downregulated FA oxidation in RCC [35,40], shedding light on potential mechanisms underlying RCC development.

In ccRCC, tumors typically display the Warburg effect, reducing glucose oxidation and TCA cycle conversion [41]. Our findings show increased l-glutamic acid but decreased l-glutamine in RCC compared to HCs, consistent with Zira et al.’s study [7]. Glutamine, a non-essential amino acid, is synthesized and metabolized in all body cells [19]. In ccRCC, glucose metabolism induced by HIF1 leads to an increase in solute carrier family 1 member 5 (SLC1A5), glucose transporter 1 or 3 (GLUT 1/3), and lactate dehydrogenase A (LDHA), thereby stimulating lactic acid production [42]. l-glutamine is absorbed through SLC1A5 and converted to l-glutamate by the enzyme glutaminase 1 or 2 (GLS1/2), entering the TCA cycle in the form of alpha-ketoglutarate (α-KG) [19]. The increased glucose in RCC enters through GLUT 1/3, leading to an elevated lactate production by the upregulated LDHA1 from pyruvate [42]. The produced lactic acid induces acidification of the tumor microenvironment, promotes inflammatory Th17 cell differentiation, and inhibits immune cells, thereby contributing to the progression of RCC [43].

In this study, l-tryptophan levels were lower in RCC than in HCs. Prior studies have shown mixed findings. Lee et al. reported elevated plasma [10], whereas Lin et al. found decreased serum [17]. RCC, marked by anorexia and negative nitrogen balance, contributes to tryptophan decline [17]. Most free tryptophan is metabolized via the kynurenine pathway, pivotal for neurochemical transmission and immune response control [44]. Enzymes, such as indoleamine-2,3-dioxygenase 1 (IDO1), IDO2, or tryptophan-2,3-dioxygenase, catalyze this, thereby modulating immunity and fostering cancer progression via tryptophan depletion and aryl hydrocarbon receptor activation [45].

In this study, the entire amino acid pathway was found to be downregulated, aligning with previous findings [6,16,46]. RCC subtypes exhibited significant changes in amino acid metabolism and redox homeostasis [46]. Glycine, serine, and threonine metabolism support synthesis and maintain redox balance through the methionine cycle [47]. Dysregulation of these pathways is linked to cellular metabolism reprogramming, supporting tumor-cell survival [48]. Our findings highlight alterations in TCA metabolism, impaired mitochondrial bioenergetics, and oxidative phosphorylation in RCC [49], underscoring the role of lipid and amino acid metabolism in carcinogenesis.

In addition, age, BMI, and gender influence plasma metabolomics and lipidomics in RCC [9]. We assessed seven major metabolites for variations by gender, age, and BMI. Older age, higher BMI, or female gender are associated with lower LysoPC levels in plasma. A large-scale study by the National Institutes of Health and the National Association for Retired Persons found being overweight increases RCC risk regardless of gender [50]. Low plasma LysoPC concentration predicts aging and indicates mitochondrial oxidative damage [51].

Numerous studies have demonstrated the influence of dietary and nutritional factors on tumor formation in patients with RCC. However, consensus regarding the significance of fat intake on renal cancer remains elusive. Our analysis using multivariate logistic regression on nutrients and food groups in RCC patients and HCs revealed that a higher intake of PUFA, *n*-3 PUFA, or *n*-6 PUFA in the overall diet was associated with an increased risk of RCC. This finding aligns with the previous results that indicated that high consumption of *n*-3 PUFA among the Japanese population, who consume a greater quantity of fish compared to Western populations, correlates with their elevated risk of RCC [29]. Conversely, individuals who consistently consumed fatty fish showed a statistically 74% lower risk of RCC [52], and there was no significant association with high levels of polyunsaturated fat, *n*-3, and n-6 PUFA intake [27]. This study is the first to investigate the correlation between metabolites and nutrition in RCC, revealing intriguing associations between PUFA intake and RCC risk.

Our study has a few limitations that need consideration. Future research should involve larger sample sizes than those used in our current study. Moreover, further confirmation through in vivo and in vitro studies could offer more precise insights into the role of the identified markers in RCC development. Additionally, more interventional studies are necessary to validate the potential impact of dietary PUFA modulation based on factors such as VHL mutation status and CPT expression profiles in RCC patients.

## 5. Conclusions

Seven potential markers for RCC diagnosis were identified and validated within plasma metabolites and lipids. Their high sensitivity and high specificity have been confirmed, rendering them valuable for diagnostic purposes. The consumption of a considerable amount of dietary PUFAs may influence the development of RCC through alterations in lipid metabolism. These findings enable efficient and accurate diagnosis of RCC and timely interventions to improve overall survival rates and promote cancer prevention through healthy lifestyle habits among individuals exposed to this cancer.

## 6. Patents

This research has led to the filing of the following patent: biomarkers for the diagnosis of renal cell carcinoma and their use (patent numbers: KR10-2023-0177712)

## Figures and Tables

**Figure 1 nutrients-16-01265-f001:**
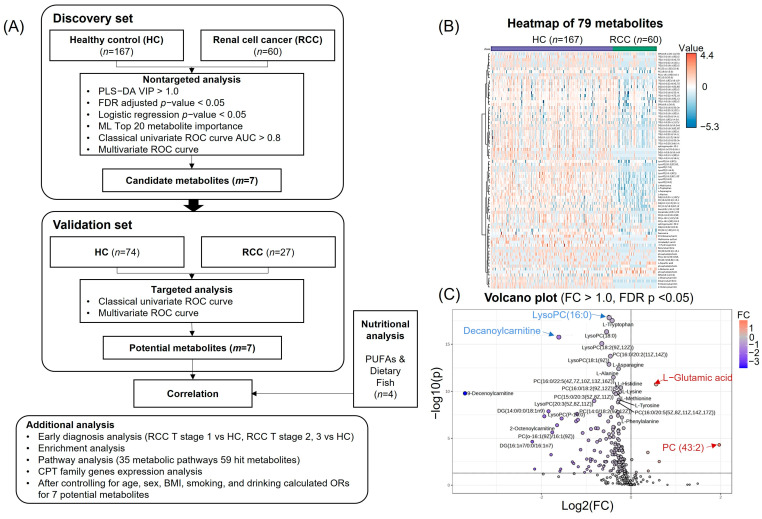
Data processing for metabolite analysis, heatmap, and volcano plot. (**A**) Flow chart of total analysis. (**B**) Heatmap of 79 metabolites between HC and RCC. (**C**) Volcano plot for HC vs. RCC. Each point in the volcano plot represents one metabolite. Significant metabolites were calculated with an FC threshold of 1.0 on the *x* axis and at FDR-adjusted *p* < 0.05 on the *y* axis. Negative log2 FC values indicated in blue represent lower concentrations in RCC than in HC (*m* = 74); positive values indicated in red represent higher concentrations of metabolites in RCC than in HC (*m* = 5).

**Figure 2 nutrients-16-01265-f002:**
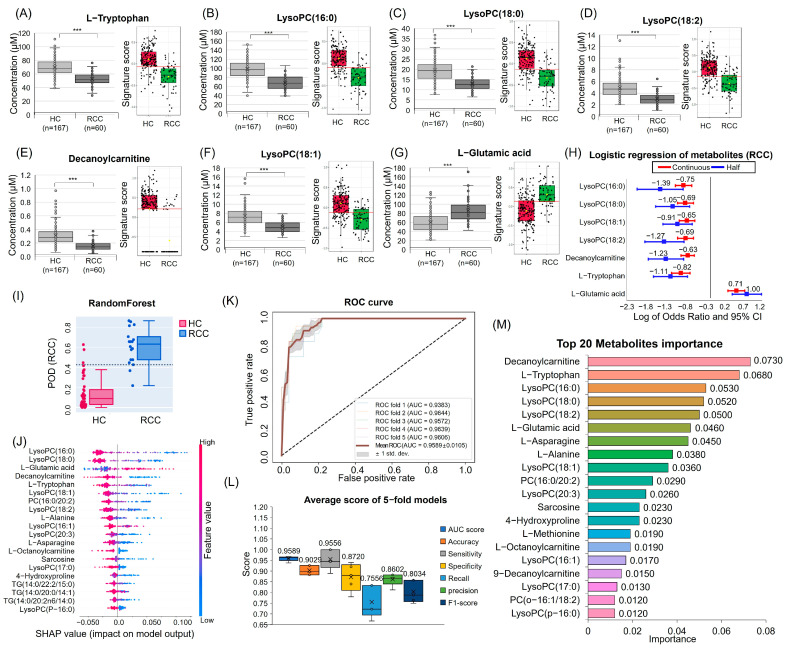
Quantitative analysis and multivariate logistic regression results of seven metabolomes and machine-learning results. Quantitative analysis and signature score results for each metabolite are (**A**) l-Tryptophan, (**B**) LysoPC (16:0), (**C**) LysoPC (18:0), (**D**) LysoPC (18:2), (**E**) Decanoylcarnitine, (**F**) LysoPC (18:1), and (**G**) l-Glutamic acid. (**H**) Multivariate logistic regression plot of ORs and 95% confidence intervals for evaluation of the relationship between HC and RCC. Adjusted for age, sex, BMI, smoking, and drinking. (**I**) POD plot (**J**) SHAP plot for one fold. (**K**) ROC curves of Random Forest. (**L**) Average score of 5-fold models. Each box plot of the AUC curve, accuracy, sensitivity, specificity, F1-score, recall, and precision was shown. (**M**) Metabolites importance plot. The top-20 important metabolites are listed and sorted by importance. *** *p* < 0.001.

**Figure 3 nutrients-16-01265-f003:**
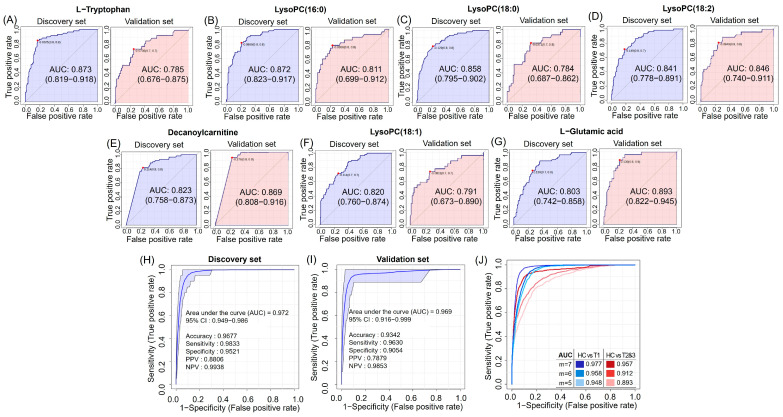
Metabolite-markers prediction of significant metabolites between HC and RCC groups. Analysis of individual metabolite markers in the discovery set and validation set: (**A**) l-Tryptophan, (**B**) LysoPC (16:0), (**C**) LysoPC (18:0), (**D**) LysoPC (18:2), (**E**) Decanoylcarnitine, (**F**) LysoPC (18:1), and (**G**) l-Glutamic acid. Multivariate ROC curve results for 7 metabolites with AUC ≥ 0.8 in (**H**) discovery set or (**I**) validation set. (**J**) Multivariate ROC curve of 5 to 7 metabolites with AUC ≥ 0.8 based on the cross-validation and the resulting HC vs. T stage 1 or T stages 2 and 3. The set of 6 metabolites excludes LysoPC (18:1) from the 7-metabolite configuration, while the 5-metabolite set further excludes l-Glutamic acid.

**Figure 4 nutrients-16-01265-f004:**
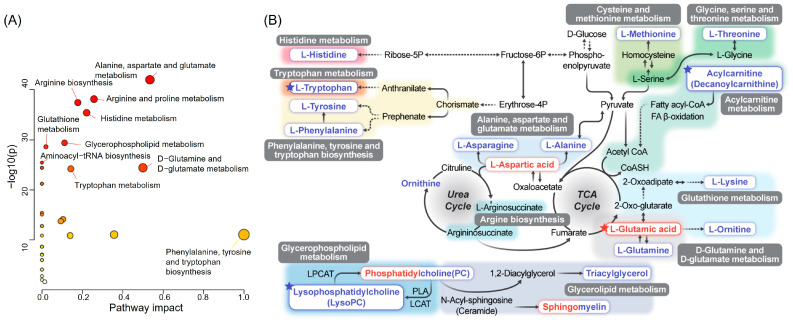
Analysis of pathway and network of differential metabolites between HC and RCC. (**A**) Overview of the pathway analysis of 79 metabolites using the MetaboAnalyst software. The color of the node reflects the *p*-value, changing from yellow to red as the significance increases, and the radius reflects the path influence value. (**B**) Major metabolites were assigned to their corresponding KEGG pathways. An increase in the relative concentration of metabolites in RCC compared to that in HC is displayed in red, whereas a decrease in relative concentration is displayed in blue. The stars indicate the major metabolites.

**Figure 5 nutrients-16-01265-f005:**
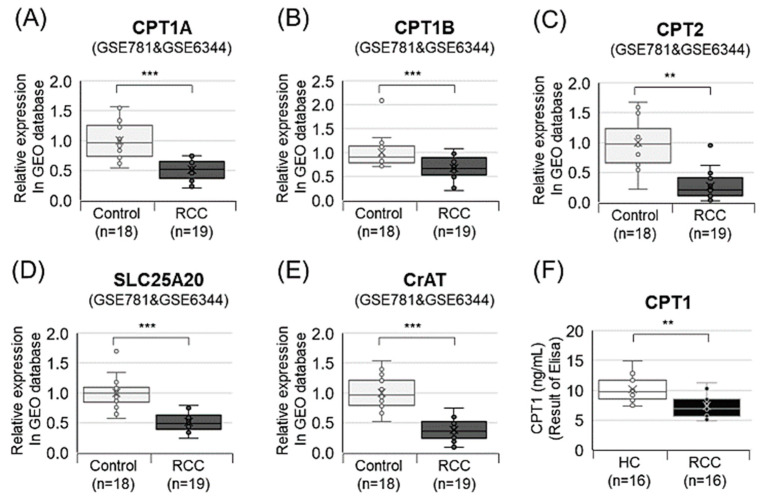
GEO database and ELISA show CPT family mRNA and levels downregulated in RCC versus controls. (**A**) CPT1A, (**B**) CPT1B, (**C**) CPT2, (**D**) SLC25A20 (CACT), and (**E**) CrAT were analyzed in the GEO database. (**F**) CPT1 was assessed using an ELISA kit. The GSE781 and GSE6344 datasets from the GEO database were used. ** *p* < 0.01; *** *p* < 0.001. CACT, carnitine-acylcarnitine translocase; CrAT, carnitine *O*-acetyltransferase.

**Figure 6 nutrients-16-01265-f006:**
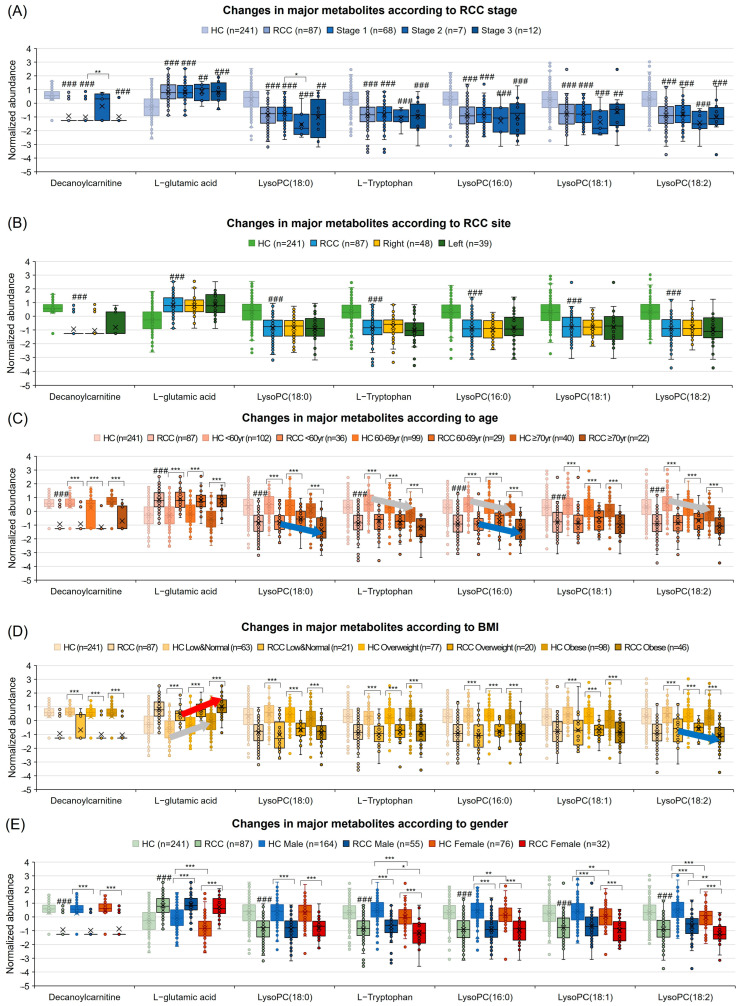
Comparison of changes according to RCC stage, site, age, BMI, and sex of seven metabolites. (**A**) RCC stage, (**B**) RCC site, (**C**) age, (D) BMI, and (**E**) sex were compared between the HC group and RCC group, while stage and site were compared within the RCC group. The diagnostic criteria for BMI were low and normal (<22.9), overweight (23.0–24.9), and obese (≥25). Significance is marked with the symbols ## *p* < 0.01 and ### *p* < 0.001 when compared to the HC group. The symbols * *p* < 0.05, ** *p* < 0.01, and *** *p* < 0.001 indicate significance between the groups. The red arrow indicates a gradual increase in the normalized abundance in the RCC group, the blue arrow indicates a gradual decrease, and the gray arrow indicates a gradual increase or decrease in the HC group.

**Figure 7 nutrients-16-01265-f007:**
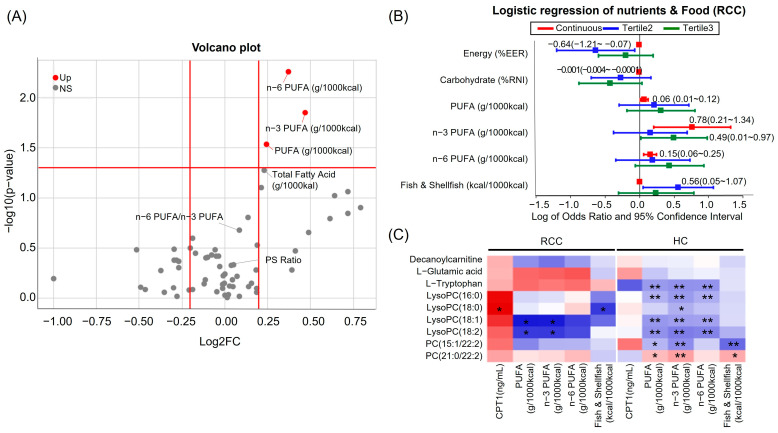
Volcano plot and analyses reveal key metabolites and fats correlation in RCC. (**A**) Volcano plot of lipid-related factors (**B**) Multivariate logistic regression plot of odds ratios and 95% confidence intervals for evaluation of the relationship between HC and RCC; (**C**) Correlation of metabolites, CPT1, PUFAs, and dietary fats in RCC. Red indicates a positive correlation, while blue indicates a negative correlation. * *p* < 0.05 and ** *p* < 0.01. PUFA; polyunsaturated fatty acid, NS; not significant, PS ratio; polyunsaturated–saturated fatty acid ratio.

**Table 1 nutrients-16-01265-t001:** General characteristics of participants according to the discovery and validation sets.

Characteristics	Discovery Set			Validation Set		
	HC (*n* = 167)	RCC (*n* = 60)	*p ^a^*	HC (*n* = 74)	RCC (*n* = 27)	*p ^a^*
Sex (Male)	121 (72.5)	35 (58.3)	0.052	44 (59.5)	20 (74.1)	0.244
Age (year)	60.0 (8.84)	62.5 (11.0)	0.109	60.8 (9.90)	60.2 (10.9)	0.759
BMI (kg/m^2^)	24.6 (2.78)	25.4 (3.99)	0.170	24.3 (3.62)	25.6 (2.41)	0.024
Low (<18.5)	3 (1.80)	2 (3.30)	0.482	3 (4.10)	0 (0.00)	0.172
Normal (18.5–22.9)	41 (24.6)	15 (25.0)		19 (25.7)	4 (14.8)	
Overweight (23.0–24.9)	51 (30.5)	13 (21.7)		26 (35.1)	7 (25.9)	
Obese (≥25)	72 (42.1)	30 (50.0)		26 (35.1)	16 (59.3)	
Experiences of smoking	86 (53.8)	26 (43.3)	0.177	40 (55.6)	16 (59.3)	0.822
Experiences of drinking	123 (79.9)	35 (58.3)	0.002	61 (83.6)	17 (63.0)	0.034

Data are presented as means (standard deviation) for continuous variables and *n* (%) for categorical variables. *^a^ p*-value calculated using the chi-square test for categorical variables and Student’s *t*-test for continuous variables. Fisher’s exact test was performed when the categorical variable was more than 25% of the cells with an expected frequency of five or less. BMI: body mass index.

## Data Availability

Study data are available from the first author (Y.-H.K.) upon request. The data are not publicly available due to privacy and ethical restrictions.

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
