# Peer review of "Influence of Dietary Polyunsaturated Fatty Acid Intake on Potential Lipid Metabolite Diagnostic Markers in Renal Cell Carcinoma: A Case-Control Study"

_nutrients, 2024, doi:10.3390/nu16091265_

Round 1
Reviewer 1 Report
Comments and Suggestions for Authors
The article submitted by Dr. Kim and collaborators represents an important contribution to the study of kidney tumors, with the proposal of lipid and non-lipid markers for diagnosis and monitoring of cases. Kidney tumors are relatively less studied than other more common tumors. The study was very well designed, with a reasonable number of participants, taking care to consider nutritional aspects and other behaviors (drinking, smoking) that could affect the results. In fact, these factors were considered in the analyses, which were very careful. The statistical analysis was very careful.
Reviewer 2 Report
Comments and Suggestions for Authors
General Comments:
The manuscript by Yeon-Hee Kim, et al conducted a thorough analysis on clinical cohorts, using a panel of metabolites to demonstrate their diagnostic power for detection of renal cell carcinoma (RCC). This study included robust wet-lab experimentation with patient samples as well as thoughtful dry-lab analysis. The design and implementation of the experiments, focusing on biomarker discovery and validation, were commendable. Additionally, the authors took great care to account for clinical variations by evaluating the diagnostic potential of these metabolites across different clinical and dietary conditions. Furthermore, they delved into the underlying mechanisms through pathway and network analysis. Overall, this manuscript would provide insightful and translational value in improvement of diagnostic performance in RCC patient. However, there are several comments and discussions that need to be addressed.
Comments:
1st, Firstly, the authors identified a set of seven metabolites as potential markers for differentiating RCC patients from healthy individuals. Figures 2 and 6 show that six out of these seven metabolites followed a similar trend of decrease in RCC patients, with the exception of L-glutamic acid. This raises a question about whether the authors intend for these metabolites to be used collectively as a panel or individually. If as a panel, is there a specific algorithm designed to measure the significance of each metabolite and produce a composite value? If the intention is to use them separately, what is the proposed method for implementation?
2nd, Regarding the potential for early detection, in Figure 3J, the authors made a comparison between T1 and T2&3. They further mention, "Higher levels of L-Glutamic acid were associated with an increased risk of RCC, ... This suggests that these seven markers could be strong candidates for RCC diagnosis, particularly in early-stage detection." It should be clarified that this analysis primarily assesses the capability of these markers to detect early stages of cancer progression, notably in metastatic scenarios, rather than estimating the overall risk of RCC. The statement should be revised accordingly.
3rd, In Figures 7 and S3, the authors illustrated the relationship between essential metabolites and dietary fats in RCC, revealing that higher intake of PUFA, n-3 PUFA, and n-6 PUFA is associated with an increased risk of RCC. This finding prompts the question of whether there are any justifications for adjusting the consumption of dietary supplements based on VHL mutation status and CPT expression profiles in individuals.
Reviewer 3 Report
Comments and Suggestions for Authors
I recommend writing a legend for the numerous abbreviations used at the beginning of the manuscript, because the large amount of information additionally loaded with numerous abbreviations makes it difficult to read.
The authors present a very interesting and up-to-date study to discover diagnostic potential markers of RCC by quantification of plasma metabolites, machine learning and marker validation. Seven potential metabolites emerged as potential markers.
I found the study very interesting, well written and illustrated.
although many data, graphs, tables, correlations are presented, they are well explained and logically arranged.
The authors indicate that the results of this study have enabled the filing of a patent application.
Comments on the Quality of English Language
Minor editing of English language required
